 

# ATRX promotes maintenance of herpes simplex virus heterochromatin during chromatin stress

Joseph M Cabral[1,2], Hyung Suk Oh[1], David M Knipe[1,2]*

[1]Department of Microbiology, Blavatnik Institute, Harvard Medical School, Boston, United States; [2]Program in Virology, Harvard Medical School, Boston, United States

**Abstract** The mechanisms by which mammalian cells recognize and epigenetically restrict viral DNA are not well defined. We used herpes simplex virus with bioorthogonally labeled genomes to detect host factors recruited to viral DNA shortly after its nuclear entry and found that the cellular IFI16, PML, and ATRX proteins colocalized with viral DNA by 15 min post infection. HSV-1 infection of ATRX-depleted fibroblasts resulted in elevated viral mRNA and accelerated viral DNA accumulation. Despite the early association of ATRX with vDNA, we found that initial viral heterochromatin formation is ATRX-independent. However, viral heterochromatin stability required ATRX from 4 to 8 hr post infection. Inhibition of transcription blocked viral chromatin loss in ATRX-knockout cells; thus, ATRX is uniquely required for heterochromatin maintenance during chromatin stress. These results argue that the initial formation and the subsequent maintenance of viral heterochromatin are separable mechanisms, a concept that likely extrapolates to host cell chromatin and viral latency.

DOI: https://doi.org/10.7554/eLife.40228.001

## Introduction

Recognition and restriction of foreign DNA is ubiquitous to all cells. It is imperative that cells prevent foreign DNA from expressing its genes or integrating into the cellular genome lest foreign gene products disrupt normal cellular function. Bacteria possess restriction endonucleases and CRISPR-Cas9 mechanisms to recognize and cleave foreign DNA (*Barrangou et al., 2007*; *Tock and Dryden, 2005*). Eukaryotic cells, however, recognize foreign DNA and redirect existing chromatin machinery to epigenetically silence the DNA by modifying it with repressive heterochromatin. Epigenetic modification of chromatin, such as DNA methylation and covalent modification of histone tails, is critical for regulating gene expression to ensure proper cellular function and expression of regulated genes at only the appropriate times (*Kanherkar et al., 2014*). Likewise, epigenetic modifications also allow for the effective silencing of foreign DNA that is often associated with invading pathogens, for example DNA viruses. DNA virus genomes are recognized by nuclear DNA sensors, such as interferon-inducible protein 16 (IFI16), and are subject to epigenetic regulation by host cell factors during both lytic and latent infection (*Knipe, 2015*; *Knipe et al., 2013*; *Orzalli and Knipe, 2014*). Nuclear unintegrated retroviral DNAs are also associated with nucleosomes and silencing histone modifications (*Wang et al., 2016*). Indeed, retroviral genomes are epigenetically silenced even after integration into the host genome (*Yao et al., 2004*).

Both boon and bane for DNA viruses, viral chromatin enables DNA viruses to establish and maintain persistent latent infections in which the virus is poised for reactivation, but it also impedes productive viral transcription and replication during lytic infection (*Knipe et al., 2013*). Viruses must overcome host-cell silencing mechanisms for reactivation from latency or for lytic infections to proceed. The double-stranded DNA genome of herpes simplex virus (HSV) is not associated with

*For correspondence:
david_knipe@hms.harvard.edu

**eLife digest** Cells carefully package their DNA, tightly wrapping the long, stringy molecule around spool-like groups of proteins called histones. However, the genes that are draped around histones are effectively silenced, because they are 'hidden' from the molecular actors that read the genetic information to create proteins. A cell can control which of its genes are active by using proteins to move histones on or off specific portions of DNA. For example, a protein known as ATRX associates with a partner to load histones onto precise DNA regions and switch them off.

Wrapping DNA around histones can also be a defense mechanism against viruses, which are tiny cellular parasites that hijack the molecular machinery of a cell to create more of themselves. For instance, the herpes simplex virus, which causes cold sores and genital herpes, injects its DNA into a cell where it is used as a template to create new viral particles. By packaging the DNA of the virus around histones, the cell ensures that this foreign genetic information cannot be used to make more invaders. However, the details of this process remain unknown. In particular, it is still unclear what happens immediately after the virus penetrates the nucleus, the compartment that shelters the DNA of the cell.

Here, Cabral et al. explored this question by dissecting the role of ATRX in silencing the genetic information of the herpes simplex virus. The viral DNA was labeled while inside the virus itself, and then tracked using microscopy imaging techniques as it made its way into the cell and inside the nucleus. This revealed that, almost immediately after the viral DNA had entered the nucleus, ATRX came in contact with the foreign molecule. One possibility was that ATRX would be responsible for loading certain forms of histones onto the viral DNA. However, after Cabral et al. deleted ATRX from the cell, histones were still present on the genetic information of the virus, but this association was less stable. This indicated that ATRX was only required to keep histones latched onto the viral DNA, but not to load the proteins in the first place.

Overall, these results show that using histones to silence viral DNA in done in several steps: first, the foreign genetic material needs to be recognized, then histones have to be attached, and finally molecular actors should be recruited to keep histones onto the DNA.

Knowing how cells ward off the herpes simplex virus could help us find ways to 'boost' this defense mechanism. Armed with this knowledge, we could also begin to understand why certain people are more likely to be infected by this virus.

DOI: https://doi.org/10.7554/eLife.40228.002

histones within the virion (*Pignatti and Cassai, 1980*). However, during lytic infection, nucleosomes are rapidly assembled on the viral DNA upon its entry to nuclei of epithelial or fibroblast cells (*Cliffe and Knipe, 2008*; *Lee et al., 2016*; *Oh and Fraser, 2008*). Newly formed viral chromatin is immediately associated with silencing histone tail modifications, specifically H3 lysine 9 trimethylation (H3K9me3) and histone H3 lysine 27 trimethylation (H3K27me3), that peak in density within 1–2 hr post infection (hpi) (*Lee et al., 2016*). The resulting viral heterochromatin acts as an epigenetic barrier to viral gene expression and replication. Consequently, viral gene expression occurs in the regulated cascade of immediate-early (IE), early (E), and late (L) viral gene products that drive viral transcription, viral DNA synthesis, and virion assembly, respectively (*Knipe and Cliffe, 2008*). HSV proteins VP16 and ICP0, a viral E3 ubiquitin ligase, promote viral gene expression, and removal of viral heterochromatin (*Cliffe and Knipe, 2008*; *Lee et al., 2016*; *Herrera and Triezenberg, 2004*). However, the cellular machinery that drives formation of viral heterochromatin and epigenetic silencing of viral gene expression during the early stages of DNA virus infections are still not well described.

The SWI/SNF chromatin remodeler protein, α-thalassemia X-linked intellectual disability (ATRX), has recently gained attention as a crucial epigenetic regulator of eukaryotic gene expression and steward of silenced heterochromatin. ATRX is also known to play a role in restriction of DNA viruses (*Lukashchuk and Everett, 2010*). ATRX and death domain-associated protein (DAXX) together form a histone chaperone complex specific for non-canonical histone variant 3.3 (H3.3) (*Drané et al., 2010*; *Lewis et al., 2010*). This complex is critical for maintaining repressive heterochromatin at many repeat-rich regions, including telomeres (*Lovejoy et al., 2012*), pericentric repeats

(*Elsässer et al., 2015*), and endogenous retroviruses (*Elsässer et al., 2015*), and mutations in the ATRX gene are linked to a developmental disorder and several cancer types (*Lovejoy et al., 2012*). ATRX and DAXX are also two of the core components of promyelocytic leukemia protein (PML) nuclear bodies (PML-NBs), nuclear punctate structures that have been implicated in a range of cellular activities including the DNA-damage response, transcriptional regulation, and restriction of viral infection (*Chang et al., 2018*). In this context, ATRX and DAXX have been shown to restrict gene expression from DNA viruses and integrated retrovirus DNA (*Lukashchuk and Everett, 2010*; *Schreiner et al., 2013*; *Shalginskikh et al., 2013*; *Tsai et al., 2011*; *Woodhall et al., 2006*). The ATRX/DAXX complex also appears to be important for maintaining DNA virus latency as depletion of either ATRX or DAXX was reported to induce EBV reactivation from latently infected cells (*Tsai et al., 2011*).

In response, DNA viruses have evolved a number of strategies for alleviating the effects of host restriction factors and epigenetic silencing. The Epstein-Barr virus (EBV) BNFR1 protein interacts with DAXX to displace ATRX from the complex, effectively reprogramming DAXX and resulting in the activation of early gene transcription (*Tsai et al., 2011*; *Tsai et al., 2014*), while degradation of DAXX is promoted by pp71 and E1B-55K during human cytomegalovirus (HCMV), and adenovirus (AdV) infections, respectively (*Schreiner et al., 2013*; *Saffert and Kalejta, 2006*). During HSV infection, proteasome-dependent degradation of PML, ATRX, and IFI16 is promoted by ICP0 (27–31). Though ATRX and DAXX have proven to be important host restriction factors, the mechanism of their antiviral activity is less clear.

While ATRX and DAXX have been shown to possess nucleosome deposition and remodeling activity in vitro (*Drané et al., 2010*; *Lewis et al., 2010*), in vivo studies have largely investigated the effects of ATRX or DAXX depletion at integrated reporter elements or viral genomes during late stage infection, after chromatin had already formed. For instance, DAXX has been reported to promote H3.3 incorporation on HCMV, AdV, and EBV genomes at 18, 24, and 72 hpi, respectively (*Schreiner et al., 2013*; *Tsai et al., 2014*; *Albright and Kalejta, 2016*). While the histone chaperones HIRA and ASF1A have been implicated in the initial loading of histones onto HSV DNA during the first few hours of infection, these reports also demonstrated that HIRA and ASF1A actually promote productive viral infection (*Oh et al., 2012*; *Placek et al., 2009*). Thus, the mechanisms underpinning de novo formation of restrictive heterochromatin on viral DNA remain unclear. Lytic HSV infection provides a temporal reference point that can be used to determine the sequential order of events and dissect the function of cellular chromatin remodelers with greater resolution. Therefore, we used HSV to investigate de novo viral heterochromatin formation and maintenance in the presence or absence of chromatin remodeler and host restriction factor ATRX.

Here we report on the mechanisms of viral gene restriction mediated by ATRX during the first 8 hours of infection. We used HSV with bioorthogonally-tagged genomes to quantitatively track viral entry and host restriction factor recruitment as a function of time. Along with nuclear DNA sensor IFI16, we detected ATRX and PML recruitment to viral DNA by 15 min post infection (mpi), almost immediately upon nuclear entry. Although ATRX restricted the expression of viral mRNA, ATRX had no effect on the initial formation of viral heterochromatin assembly at 2 and 4 hpi. However, ATRX was required for the maintenance of viral heterochromatin between 4–8 hpi during challenges to chromatin stability. Our findings argue for a biphasic model of epigenetic regulation in which de novo assembly of heterochromatin on viral genomes is ATRX-independent but that ATRX is required for viral heterochromatin stability during chromatin stress, such as replication and transcription.

## Results

### HSV genomes colocalize with host restriction factors IFI16, ATRX, and PML upon entry into the nucleus

To investigate early interactions of HSV genomes with host proteins that could potentially initiate chromatinization of viral DNA, we generated nucleoside analog-labeled HSV-1 preparations that could be used for the detection of input viral DNA in infected cells. First, we prepared labeled HSV (HSV-EdC) stocks by infecting confluent human foreskin fibroblasts (HFF) with HSV-1 in the presence of the nucleoside analog 5-ethynyl-2′-deoxycytidine (EdC, 0.5 μM). HSV-EdC DNA could be detected by fluorescence microscopy after a copper-catalyzed bioorthogonal cycloaddition (click

chemistry) of biotin to EdC incorporated into viral DNA followed by incubation with a streptavidin-fluorophore probe (*Figure 1A*; *Figure 1—figure supplement 1B* ). HSV-EdC genomes were detected as punctate nuclear foci that colocalized with the HSV immediate-early protein ICP4, considered a marker for viral DNA, at 2 hr post infection (hpi) (*Figure 1—figure supplement 2A*). We also observed that HSV-EdC viral DNA (vDNA) colocalized with early replication compartments, as defined by immunofluorescence (IF) staining for the viral early protein ICP8 at 4 hpi (*Figure 1—figure supplement 2B*). As recently reported (*Sekine et al., 2017*), HSV-EdC genomes that colocalized with ICP8 underwent a morphological decompaction that could be prevented by treatment with viral DNA synthesis inhibitor phosphonoacetic acid (PAA, 400 µg/ml) (*Figure 1—figure supplement 2B*). HSV-EdC demonstrated only a slight decrease in ICP8 protein levels at 4hpi but no differences in either ICP4 or ICP8 levels by 8hpi (*Figure 1—figure supplement 2C*). Intracellular HSV-EdC foci were prevented by treatment with heparin (50 µg/mL), which blocks viral attachment and entry to cells (*Bender et al., 2005*) (*Figure 1—figure supplement 2D*). Labeling of host cell chromatin during infection with EdC-labeled viral preparations was not observed. In total, these results validated the use of EdC labeling of HSV to detect input HSV DNA and showed that the course of infection with HSV-EdC virus was normal.

To determine the kinetics of restrictive host protein recruitment to input viral DNA, we quantitatively mapped the spatiotemporal kinetics of incoming HSV-EdC genome colocalization with host restriction factors IFI16, PML, and ATRX (*Figure 1A,C–E*; *Figure 1—figure supplement 3A,B* ; *Video 1*). To measure colocalization of viral DNA and restriction factor foci in host cell nuclei, we first defined nuclear fluorescent foci using a software bot (*Cicconet et al., 2017*) and then used additional customized features of the software to define foci as colocalizing when the centers were within a distance threshold of 5 pixels,~350 nm (see Materials and methods for detailed description of software). Using this method, we observed click chemistry labeled viral genomes in the nuclei of HFFs as early as 15 mpi (*Figure 1B*). The percentage of infected cells increased steadily over the first 30 min of infection. A stable 2–3 genomes per infected nucleus were maintained from 50 mpi (*Figure 1—figure supplement 3C*). ATRX colocalization with HSV DNA peaked between 40–100 mpi (*Figure 1E*), with 80% of viral DNA colocalizing with ATRX. We observed IFI16 and ATRX colocalizing with viral DNA at similar frequencies by 15 mpi (*Figure 1C*). IFI16 has been shown to form transient foci at the nuclear periphery in response to HSV infection (*Diner et al., 2016*; *Everett, 2016*). The kinetics of IFI16 colocalization with viral DNA supported this hypothesis as our observations revealed an early peak of IFI16-vDNA colocalization (20–30 mpi) that rapidly decreased after 30 mpi (*Figure 1C,E*). Little to no IFI16-vDNA colocalization was observed from 60 mpi (*Figure 1E*). These results indicated that, under our conditions, IFI16 and ATRX localize with HSV genomes almost immediately upon nuclear entry, and while the majority of IFI16 colocalization with input viral DNA occurs between 15 and 30 mpi, ATRX colocalization is stable between 15 and 100 mpi.

PML and ATRX colocalization with viral DNA occurred with nearly identical kinetics (*Figure 1D*). We attempted to analyze the kinetics of DAXX colocalization with HSV DNA. However, several antibodies, while providing a signal for immunoblotting, gave little or no signal in IF experiments. ATRX colocalization with viral DNA began to wane by 100 mpi (*Figure 1E*). The HSV E3 ubiquitin ligase, ICP0, which has been shown to promote degradation of both PML and ATRX (*Chelbi-Alix and de Thé, 1999*; *Jurak et al., 2012*). Indeed, we observed that the presence of ATRX foci in infected nuclei generally diminished as the presence of ICP0 increased across the infected population (*Figure 1—figure supplement 3D*). By 90 min post infection, ICP0 was observed to localize to ATRX foci that also colocalized with HSV genomes (*Figure 1—figure supplement 3E*). Though the percentage of infected cells positive for ICP0 did not change between 90 and 120 min, the percentage of infected cells positive for ATRX foci decreased by more than half (*Figure 1—figure supplement 3D*). This may reflect a period of time in which ICP0 drives the dissolution of PML-NBs; however, the ICP0 antibody exhibited a high background signal that rendered it difficult to perform precise colocalization studies that would be needed to further define the spatiotemporal relationship between ICP0 and ATRX.

To test whether the colocalization of vDNA with ATRX or IFI16 was due to random events, we compared the distances of vDNA to restriction factors with the distances of random intranuclear points to vDNA. Frequency distribution analysis of pixel distances from HSV-EdC genomes to nearest-neighbor (nn) ATRX foci revealed a non-parametric distribution of distances that had a median distance of 2.83 pixels (~200 nm), within the optical resolution of the system (*Figure 1F*). Similarly,

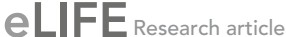

**Figure 1.** HSV genomes colocalize with host restriction factors upon nuclear entry. HFFs were infected with HSV-EdC at an MOI of 5 and fixed at times post infection as indicated. Host proteins were detected by immunocyctochemistry, and HSV genomes were detected with click chemistry and a streptavidin-fluorophore probe. (A) Representative confocal image showing colocalization of ATRX and IFI16 with vDNA as indicated by the white arrows. (B) Percentage of cells with at least 1 HSV-EdC focus within the nucleus as determined by foci detection software. n ≥ 270 total cell population

*Figure 1 continued on next page*

*Figure 1 continued*

per time point from 15 to 30 mpi derived from 3 independent experiments. Percentage of vDNA foci colocalized with (C) ATRX (blue) or IFI16 (black) and (D) ATRX (blue) or PML (black) from 15 to 30 mpi in 5 min intervals as determined by foci detection and colocalization software (see Materials and methods). n ≥ 150 total cell population per time point from 15 to 30 mpi derived from 3 independent experiments. (E) Percentage of vDNA foci colocalized with ATRX (blue) or IFI16 (black) from 30 to 120 mpi in 10 min intervals as determined by foci detection and colocalization software. n ≥ 65 total cell population per time point from 30 to 120 mpi derived from 3 independent experiments. (F) Frequency distribution of distances from the center of a vDNA focus to the center of the nearest neighbor ATRX focus within a 60 pixel radius of the vDNA focus (blue) vs a distribution of distances from vDNA to randomly generated points within the nucleus equal to the number of ATRX foci (see *Figure 1—figure supplement 1A*). Distance values were binned in increments of 1 pixel with the center of the first bin set to 0.5 pixels. Kolmogorov-Smirnov test was used to compare distributions and generate a p value. 0 = same underlying distribution, 1 = d distributions are significantly different (see Materials and methods). n = 183 vDNA foci. (G) Frequency distribution of distances (as in F) from the center of a vDNA focus to the center of an IFI16 focus vs vDNA to random points. n = 51 vDNA foci. All data are reported ± standard error of the mean.

DOI: https://doi.org/10.7554/eLife.40228.003

The following figure supplements are available for figure 1:

**Figure supplement 1.** Foci and colocalization detection.

DOI: https://doi.org/10.7554/eLife.40228.004

**Figure supplement 2.** EdC labeling of KOS HSV.

DOI: https://doi.org/10.7554/eLife.40228.005

**Figure supplement 3.** HSV DNA colocalization with ATRX, IFI16, and PML.

DOI: https://doi.org/10.7554/eLife.40228.006

vDNA-to-nnIFI16 foci had a mean median distance of 2.24 pixels (~150 nm) (*Figure 1G*). We used a custom software script to generate foci at randomly positioned x,y coordinates in numbers equal to the ATRX (or IFI16) foci that fell within a given nucleus (see Materials and methods for detailed description of software). We then compared the frequency distribution of vDNA-to-nnATRX foci (or vDNA-to-nnIFI16 foci) distances with the distribution of distances of vDNA to randomly generated x, y coordinates (*Figure 1—figure supplement 1A*). The frequency distributions of distances from vDNA-to-nnATRX foci and vDNA-to-nnIFI16 foci versus the distribution of distances from vDNA-to-nnRandom points at 30mpi were highly significant (p < 0.001) (*Figure 1F,G*). These results argued that IFI16 and PML-NB proteins localize to viral DNA very early and are candidates for the DNA sensors that initially respond to nuclear entry of viral genomes.

## ATRX colocalization with HSV genomes is independent of IFI16 but dependent on the histone chaperone DAXX

We next investigated if IFI16 affects ATRX colocalization with HSV-EdC genomes or vice versa. We reported that IFI16 and PML recruitment to viral genome complexes are independent (*Orzalli et al., 2013*), while another study observed only a 10% reduction in cells showing PML recruitment to genome complexes when IFI16 was depleted (*Cuchet-Lourenço et al., 2013*). We depleted HFFs of IFI16 using small interfering RNAs (siRNA) against IFI16 (siIFI16) (*Figure 2A,C*). Cells treated with siIFI16 and infected at a multiplicity of infection (MOI) of 5 with HSV-EdC showed no significant reduction in ATRX colocalization with viral genomes at 30mpi (*Figure 2A,D*). Likewise, depletion of ATRX via siRNA (siATRX) did not change the colocalization frequency of IFI16 with labeled HSV genomes (*Figure 2B,E*). Interestingly, treatment with siRNAs reduced the frequency of IFI16 foci formation in both siNon-Targeting (siNT) and siATRX treated HFFs (*Figure 2E*), perhaps due to an innate immune response to transfected

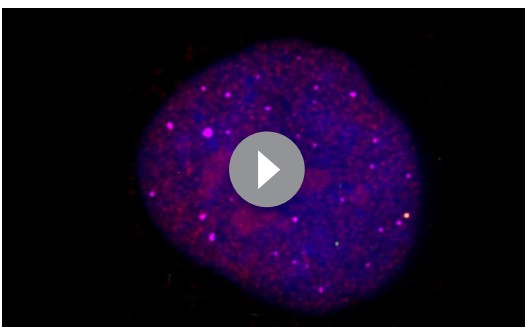

**Video 1.** 3D rendering of HSV-EdC colocalization with ATRX and IFI16 at 30 min post infection. 3D rendering from confocal imaging data captured from HFFs infected with HSV-EdC at 30 min post infection. HSV DNA is shown in green; ATRX is shown in magenta; IFI16 is shown in red; DAPI is shown in blue.

DOI: https://doi.org/10.7554/eLife.40228.007

**Figure 2.** ATRX and IFI16 independently localize to viral DNA. HFFs were treated with siRNAs targeting ATRX (siATRX), IFI16 (siIFI16), DAXX (siDAXX), or non-targeting (siNT). 72 hr post siRNA treatment, cells were infected with HSV-EdC at an MOI of 5. (**A**) Colocalization of ATRX (magenta) and IFI16 (red) with vDNA (green) in HSV-EdC infected HFFs treated with either siNT or siIFI16 at one hpi. (**B**) Colocalization of ATRX and IFI16 with vDNA in HSV-EdC infected HFFs treated with siATRX at 30 mpi. (**C**) Immunoblot detection of ATRX and IFI16 in lysates from HFFs treated with siRNA against either

*Figure 2 continued on next page*

*Figure 2 continued*

ATRX or IFI16 72 hr post treatment. GAPDH was used as a loading control. Percentage of vDNA that colocalizes with (**D**) ATRX or (**E**) IFI16 in cells treated with siNT, siATRX, or siIFI16 30 mpi as determined by foci detection and colocalization software. n ≥ 120 total cell population derived from two independent experiments, one-way ANOVA. All data are reported ± standard error of the mean. (**F**) Colocalization of ATRX and IFI16 with vDNA in HSV-EdC infected HFFs treated with siDAXX at 30 mpi.

DOI: https://doi.org/10.7554/eLife.40228.008

The following figure supplement is available for figure 2:

**Figure supplement 1.** siRNA depletion of ATRX and DAXX in HFF cells.

DOI: https://doi.org/10.7554/eLife.40228.009

siRNA (*Whitehead et al., 2011*). While ATRX recruitment to viral DNA appears to be independent of IFI16, it has been reported that mutation or alteration of the ATRX interaction domain of DAXX abolished ATRX localization to PML-NBs and early replication compartments (*Lukashchuk and Everett, 2010*). Indeed, ATRX colocalization with vDNA and PML-NBs was lost in HFFs depleted for DAXX, resulting in diffuse nuclear ATRX staining (*Figure 2F*, *Figure 2—figure supplement 1A* ). These results demonstrated IFI16 is not required for ATRX colocalization to viral DNA, as recently reported for PML and IFI16 (*Alandijany et al., 2018*), and argued that ATRX and IFI16 are in distinct DNA sensing pathways that simultaneously detect input viral DNA as it enters the nucleus.

## ATRX and DAXX act together to restrict HSV gene expression via an IFI16-independent pathway

We next investigated the functional effects of ATRX on epigenetic silencing of viral DNA. Depletion of ATRX from HepaRG cells was reported to increase viral plaque formation and slightly increase viral early protein UL42 as detected by immunoblotting (*Lukashchuk and Everett, 2010*). Because ICP0 promotes the dissolution of PML-NBs and the proteasome-mediated degradation of ATRX, we performed the following studies using the HSV-1 7134 virus that is ICP0-null.

ATRX colocalized with input viral DNA from 15 to 100 mpi; therefore, we investigated if ATRX was associating with the viral DNA. We performed chromatin immunoprecipitation followed by quantitative polymerase chain reaction (ChIP-qPCR) on chromatin from HFF cells infected with HSV 7134 virus. We found that ATRX could be detected at viral gene promoters for both *ICP27* and *ICP8* at levels higher than GAPDH by one hpi, and to significantly higher levels by 4 hpi (*Figure 3A*). Detection of ATRX at viral gene promoters suggested that ATRX may play a role in epigenetically regulating viral gene expression by associating with viral DNA.

We next measured viral gene expression in siATRX-treated HFFs infected with HSV 7134. We harvested infected cells at 2 hr intervals from 2 to 8 hpi and measured viral transcripts by reverse transcription (RT) -qPCR (*Figure 3B–D*). ATRX-depleted HFFs showed significant increases in transcripts from genes of all kinetic classes, with the most significant effects on expression occurring from IE (*ICP27*) and L genes (*gB*) at 6 and 8 hpi (*Figure 3B,D*), while E gene *ICP8* expression was significantly elevated at 8hpi (*Figure 3C*). In parallel with the above experiment, we tested the impact of viral DNA replication on ICP0-null HSV gene expression in HFFs depleted of ATRX. To accomplish this, we treated cells with a viral DNA polymerase inhibitor, PAA (400 µg/ml), from 1 hr prior to infection and maintained PAA throughout the experiment. While overall viral gene expression was reduced in the presence of PAA, depletion of ATRX still resulted in significant increases in ICP0-null gene expression from each gene of the three kinetic classes (*Figure 3B–D*). The increased accumulation of viral mRNA upon ATRX depletion argued that ATRX plays a role in preventing transcription from viral genes, and the increase in viral gene expression with and without PAA demonstrated that ATRX restricts gene expression from both input and progeny viral DNA.

To facilitate our functional studies of ATRX and DAXX, we used CRISPR-Cas9 mediated gene editing to establish an ATRX-knockout cell line (ATRX-KO) derived from hTERT immortalized human fibroblasts (*Albright and Kalejta, 2016*; *Bresnahan and Shenk, 2000*). We also established a control cell line (Control) in parallel that expresses Cas9 but no guide RNA, resulting in passage-matched ATRX-KO and Control cell lines (*Figure 4—figure supplement 1A* ). The immortalized fibroblasts were not permissive for single cell cloning; therefore, we used a population of ATRX-KO cells maintained under puromycin selection. ATRX-KO cells yielded significantly higher viral titers of ICP0-null virus than Control cells (MOI 3) (*Figure 4—figure supplement 1B* ). Similar to our observations in



**Figure 3.** ATRX restricts HSV gene expression from input and progeny viral DNA. (**A**) HFFs were infected with HSV 7134 at an MOI of 3, and infected cells were fixed and harvested 30, 60, and 240 min post infection. ChIP-qCPR and HSV specific primers were used to detect chromatin enrichment of ATRX at ICP27 (blue) and ICP8 (black) gene promoters. Two-tailed t-tests were used to compare ATRX enrichment at viral gene promoters compared to

*Figure 3 continued*

GAPDH. (B) HFFs were treated with siNT or siATRX and infected with HSV 7134 at an MOI of 5 in the absence (left panels) or presence (right panels) of PAA. Relative viral transcripts for (B) *ICP27*, (C) *ICP8*, or (D) *gB* were quantified by qPCR at 0, 2, 4, 6, and 8 hpi. Viral mRNA levels were normalized to cellular 18S transcripts. Results were analyzed by two-way ANOVA. All data for *Figure 3* are reported as the average of 3 independent experiments ± standard error of the mean; p < 0.05 (*), p < 0.01 (**), p < 0.001 (***).
DOI: https://doi.org/10.7554/eLife.40228.010

siRNA treated cells, *ICP27* gene expression from ICP0-null HSV was significantly greater in ATRX-KO cells than Control by 4 hpi, with both *ICP8* and *gB* exhibiting significantly elevated expression levels by six hpi (*Figure 4—figure supplement 1C*).

DAXX has also been shown to reduce HSV U$_L$42 protein levels during ICP0-null HSV infection (*Lukashchuk and Everett, 2010*); however, the effects of double depletion of ATRX and DAXX on viral mRNA levels have yet to be investigated. We treated ATRX-KO and Control cells with siRNAs against DAXX, ATRX, or a non-targeting control (*Figure 4—figure supplement 1D*). Control cells treated with siDAXX exhibited elevated expression of *ICP27* transcripts and a slight increase in *ICP8* transcript levels by 8 hpi (*Figure 4A*). In contrast, siDAXX-treated ATRX-KO cells did not exhibit elevated expression of either *ICP27* or *ICP8* in comparison to siNT and siATRX treated ATRX-KO at 8 hpi (*Figure 4A*). Slightly elevated viral gene expression from ATRX-KO cells treated with siATRX is likely due to increased depletion efficiency in the heterogeneous ATRX-KO population. These results argued that DAXX works in conjunction with ATRX to restrict viral gene expression. Interestingly, siDAXX treated ATRX-KO cells exhibited a significant decrease in *gB* transcripts at 8 hpi when compared to siATRX treated ATRX-KO cells (*Figure 4A*). Similarly, Control cells treated with siDAXX did not exhibit an increase in *gB* expression at 8 hpi. These results indicated that DAXX restricts viral gene expression while in complex with ATRX, but after ATRX is degraded by the activity of ICP0 (30), it may have a separate role in promoting *gB* and potentially other viral late gene expression or replication.

We next investigated the effects of DAXX depletion in ATRX-KO cells on viral yield. ATRX-KO cells treated with siDAXX and infected with ICP0-null HSV at either MOI 0.1 or 3 exhibited no increase in viral yield in comparison to cells treated with siATRX and siATRX + siDAXX (*Figure 4B*, *Figure 4—figure supplement 1E*). Treatment with siNT decreased viral yield in both ATRX-KO and Control cells compared to untreated samples, possibly due to the cellular response to siRNA transfection as discussed above (*Figure 4—figure supplement 1E*). Unlike siDAXX treatment in ATRX-KO cells, siIFI16 and siATRX treatment resulted in a significant increase in viral yield over siATRX treatment alone in ATRX-KO cells (*Figure 4C*). We recently reported similar additive effects on ICP0-null HSV yield in cells depleted for both DAXX and IFI16 (*Merkl et al., 2018*). These results supported a model in which ATRX restricts HSV gene expression in coordination with DAXX, and this pathway is distinct from IFI16 sensing and restriction of viral DNA.

## ATRX depletion increases ICP0-null HSV DNA replication and removal of heterochromatin

ATRX and DAXX have been reported to promote H3.3 deposition on viral DNA promoters of AdV and HCMV (*Schreiner et al., 2013*; *Albright and Kalejta, 2016*; *Newhart et al., 2013*); however, the role(s) that ATRX and DAXX play in chromatin deposition on HSV DNA remain unknown. To investigate if ATRX plays a role in the initial chromatin deposition on input viral DNA, we performed ChIP-qPCR. ATRX-KO and Control cells were infected with ICP0-null 7134 virus at an MOI of 3, and cell lysates were harvested at 2, 4, and 8 hpi. Chromatin was immunoprecipitated with antibodies specific for total H3, H3.3, H3K9me3, and H3K27me3. We then quantified the recovered chromatin via qPCR. Surprisingly, we detected little to no difference in chromatin occupation at either the *ICP27* or *ICP8* gene promoter at 2 and 4 hpi for each of the chromatin markers tested in ATRX-KO and Control cells (*Figure 5A–D*; *Figure 5—figure supplement 1A-D*). However, by 8 hpi, we observed lower levels of viral chromatin in ATRX-KO cells, with H3, H3.3, H3K9me3, and H3K27me3 all exhibiting significantly decreased levels compared to Control cells at both *ICP27* and *ICP8* gene promoters (*Figure 5A–D*; *Figure 5—figure supplement 1A-D*). The overall density of H3K9me3 modifications per H3 at 8 hpi in ATRX-KO cells was less than half of those in Control cells

**Figure 4.** ATRX and DAXX cooperatively restrict HSV gene expression via an IFI16-independent pathway. (**A**) Relative viral transcripts for *ICP27*, *ICP8*, and *gB* detected by qPCR in whole cell lysates collected from ATRX-KO or Control cells treated with siNT, siDAXX, or siATRX for 72 hr then infected with HSV 7134 at an MOI of 5. Lysates were collected at 4 and 8 hpi. Results were analyzed by two-way ANOVA. (**B**) Viral yields from ATRX-KO and Control cells treated with siRNAs against non-targeting, ATRX, DAXX, or ATRX +DAXX (siD +A) and infected with HSV 7134 at an MOI of 0.1. Viral lysates were collected at 24 hpi and titrated on U2OS cells. Yields were normalized to (PFU/mL)/1 × 10$^5$ cells. Results were analyzed by two-way ANOVA. (**C**) Viral yields from ATRX-KO and Control cells treated with siRNAs against non-targeting, ATRX, IFI16, or ATRX + IFI16 (siA + I) and infected with HSV 7134 at an MOI of 0.1. Viral lysates were collected at 48 hpi and titrated on U2OS cells. Yields were normalized to (PFU/mL)/1 × 10$^5$ cells. Results were analyzed by two-way ANOVA. All data for *Figure 4* are reported as the average of 3 independent experiments ± standard error of the mean; $p < 0.05$ (*), $p < 0.01$ (**), $p < 0.001$ (***).

DOI: https://doi.org/10.7554/eLife.40228.011

The following figure supplement is available for figure 4:

**Figure supplement 1.** CRISPR-mediated knockout of ATRX alleviates viral restriction.

DOI: https://doi.org/10.7554/eLife.40228.012

---

(*Figure 5E*). This argued that loss of H3K9me3 occurred at a greater rate than the removal of H3. Indeed, H3K9me3 levels were reduced by greater than 10-fold at the *ICP27* promoter in ATRX-KO cells compared to Control. DNA replication was also enhanced in ATRX-KO cells, indicating an earlier onset of viral DNA synthesis (*Figure 5F*). These findings were supported by the observation that when infected with an ICP0-positive virus, 7134R, there was no significant difference in viral gene expression between ATRX-KO and Control cells (*Figure 5—figure supplement 1F*). This argued

**Figure 5.** ATRX depletion enhances ICP0-null HSV DNA Replication and removal of heterochromatin. ATRX-KO or Control cells were infected with HSV 7134 at an MOI of 3. Infected cells were fixed and harvested 2, 4, and 8 hpi. ChIP-qCPR and HSV specific primers were used to detect chromatin

*Figure 5 continued on next page*

*Figure 5 continued*

enrichment of (**A**) H3, (**B**) H3.3, (**C**) H3K9me3, and (**D**) H3K27me3 at the viral gene promoter for *ICP27*. Results are reported as the percent of input immunoprecipitated by each antibody. Two-tailed t-tests were used to compare results from ATRX-KO versus Control cells for each antibody and each time point. (**E**) H3K9me3 enrichment per H3 for the *ICP27* promoter. (**F**) Chromatin input for *ICP8* relative to input *GAPDH* to determine relative viral genome copy numbers. All data for *Figure 5* are reported as the average of 4 independent experiments ± standard error of the mean; p < 0.05 (*), p < 0.01 (**).

DOI: https://doi.org/10.7554/eLife.40228.013

The following figure supplement is available for figure 5:

**Figure supplement 1.** ATRX depletion enhances removal of heterochromatin from the ICP8 promoter.

DOI: https://doi.org/10.7554/eLife.40228.014

that the unique restrictive effects exerted by ATRX on ICP0-null HSV gene expression occurred later than 2 hpi, when ICP0 disrupted PML-NBs (*Figure 1—figure supplement 3D*), possibly in response to chromatin stability challenges, such as replication and transcription. From these results we concluded that ATRX promotes maintenance of stable viral heterochromatin during infection and delays the onset of viral DNA replication but is not uniquely required for the de novo formation of chromatin on input viral genomes.

## ATRX promotes accumulation of heterochromatin on input viral genomes

To determine if the changes in heterochromatin composition on ICP0-null HSV genomes were dependent on viral DNA synthesis, we performed ChIP-qPCR in ATRX-KO and Control cells with or without viral DNA synthesis inhibitor. ATRX-KO and Control cells were pretreated with PAA for 1 hr prior to infection and maintained for the duration of the experiment. We infected cells with HSV 7134, harvested at 4 and 8 hpi, and performed ChIP-qPCR as described above. Consistent with our previous results, untreated ATRX-KO cells exhibited reduced H3, H3.3. H3K9me3, and H3K27me3 at both the *ICP27* and *ICP8* promoters compared to their Control cell counterparts (*Figure 6—figure supplement 1A-D* and *2A-D* ). However, when viral DNA synthesis was inhibited with PAA (*Figure 6E*), viral chromatin density was similar in all conditions at 4 hpi (*Figure 6A–D*). Viral heterochromatin continued to accumulate from 4 to 8 hpi in PAA-treated Control cells while it remained near 4 hr levels in PAA treated ATRX-KO cells; however, removal of heterochromatin was largely blocked by PAA treatment (*Figure 6A–D*). Interestingly, H3.3 increased in occupancy at the *ICP8* promoter from 4 to 8 hpi in ATRX-KO cells treated with PAA suggesting that H3.3 deposition may be differentially regulated at the *ICP8* promoter (*Figure 6—figure supplement 2C* ). Overall, these results argued that ATRX maintained virus genome-associated heterochromatin on input HSV DNA, promoted increased viral heterochromatin stability and density from 4 to 8 hpi, and that removal of heterochromatin was enhanced by viral DNA replication.

## Transcription contributes to HSV heterochromatin instability in the absence of ATRX

Chromatin structure is dynamic by nature and requires active maintenance in response to stability challenges including replication and transcription (*Schwabish and Struhl, 2004*; *Nair et al., 2017*). To assess the effects of transcription on viral chromatin composition, we infected Control and ATRX-KO cells with transcriptionally silent HSV *d*109 virus. ATRX-KO cells showed no difference in either H3 or H3K9me3 levels at the *ICP8* gene promoter at both 4 and 8 hpi (*Figure 7A*). To further investigate the effects of ATRX on chromatin during transcription, we inhibited transcription with CDK9 inhibitor flavopiridol. Cells were treated with flavopiridol (1 µM, 1 hr pretreatment) (*Figure 7—figure supplement 2E*) and infected with HSV. As previously observed in untreated cells, H3 and H3K9me3 levels on HSV DNA were similar at 4 hpi in ATRX-KO and Control cells but lower by 8 hpi in ATRX-KO fibroblasts ( *Figure 7—figure supplement 1A,B* ). However, treatment with flavopiridol blocked heterochromatin loss and resulted in H3 and H3K9me3 accumulation from 4 to 8 hpi in ATRX-KO cells (*Figure 7B*, *Figure 7—figure supplement 2A* ). Likewise, treatment with polymerase inhibitor α-amanitin (2 µg/ml, 16 hr pretreatment) (*Figure 7—figure supplement 2F*) blocked removal of H3 and H3K9me3 in ATRX-KO cells and resulted in heterochromatin accumulation from 4 to 8 hpi (*Figure 7C*, *Figure 7—figure supplement 1C,D* and *2B* ). We also infected Control and ATRX-KO



**Figure 6.** ATRX promotes maintenance of heterochromatin on input viral genomes. ATRX-KO or Control cells were infected with HSV 7134 at an MOI of 3. Infected cells were fixed and harvested 8 hpi. ChIP-qCPR and HSV specific primers were used to detect chromatin enrichment of at the viral gene promoters in the absence (*Figure 6—figure supplement 1A–D*) or presence of PAA. Enrichment of (A) H3 and (B) H3K9me3 at the *ICP27* gene promoter. Enrichment of (C) H3 and (D) H3K9me3 at the *ICP8* gene promoter. Results reported as Relative IP (the percent of input immunoprecipitated by each antibody normalized to the 4 hr control sample - set to 1.0 for each replicate). (E) Chromatin input for *ICP8* relative to input *GAPDH* to determine relative viral genome copy numbers in the absence or presence of PAA. All data for *Figure 6* are reported as the average of 4 independent experiments ± standard error of the mean; two-tailed t-test, $p < 0.05$ (*), $p < 0.01$ (**).

DOI: https://doi.org/10.7554/eLife.40228.015

The following figure supplements are available for figure 6:

**Figure supplement 1.** Untreated controls for ChIP.
DOI: https://doi.org/10.7554/eLife.40228.016

**Figure supplement 2.** ATRX promotes maintenance of H3K27me3 and H3.3 on input viral genomes.
DOI: https://doi.org/10.7554/eLife.40228.017

**Figure 7.** ATRX promotes heterochromatin maintenance during challenges to chromatin stability. (**A**) ATRX-KO and Control cells were infected with HSV *d*109 at an MOI of 3 and fixed and harvested at 4 and 8 hpi. ChIP-qPCR was used to detect H3 and H3K9me3 enrichment at the *ICP8* promoter. (**B**) ATRX-KO and Control cells were treated with flavopiridol (flavopiridol; 1 μM) from 1 hr prior to infection until time of harvest. Treated and untreated (*Figure 7—figure supplement 1A–B*) cells were infected with HSV 7134 at MOI three and fixed at 4 and 8 hpi. ChIP-qPCR was used to detect H3 and H3K9me3 enrichment at the *ICP8* promoter. (**C**) ATRX-KO and Control cells were treated with α-amanitin (2 μg/mL) from 16 hr prior to infection until time of harvest. Treated and untreated (*Figure 7—figure supplement 1C–D*) cells were infected with HSV 7134 at MOI three and fixed at 4 and 8 hpi. ChIP-qPCR was used to detect H3 and H3K9me3 enrichment at the *ICP8* promoter. (**D**) ATRX-KO and Control cells were infected with HSV 7134 at a MOI of 3 and treated with ActD (5 μg/mL) at 4 hpi. Samples were fixed and collected a 4 and 8 hpi. Chromatin enrichment is reported as percent input immunoprecipitated by antibodies specific for H3 and H3K9me3 as detected by qPCR using specific primers for the promoter of *ICP8*. All data for *Figure 7* are reported as the percent of input immunoprecipitated by each antibody and are the average of 3 independent experiments ± standard error of the mean; two-tailed t-test, $p < 0.05$ (*).

DOI: https://doi.org/10.7554/eLife.40228.018

*Figure 7 continued on next page*

*Figure 7 continued*

The following figure supplements are available for figure 7:

**Figure supplement 1.** Untreated controls for flavopiridol and α-amanitin ChIPs.
DOI: https://doi.org/10.7554/eLife.40228.019
**Figure supplement 2.** ATRX promotes heterochromatin maintenance at the ICP27 promoter during chromatin stress.
DOI: https://doi.org/10.7554/eLife.40228.020
**Figure supplement 3.** Relative HSV genomes during drug treatment.
DOI: https://doi.org/10.7554/eLife.40228.021

cells with HSV 7134 and added the transcriptional inhibitor actinomycin-D (actD, 5 µg/ml) at 4 hpi. Addition of actD prior to infection blocked chromatin loading to HSV DNA (not shown), so we added actD at 4 hpi to arrest transcription once viral chromatin had been established. Treatment with actD at 4 hpi stabilized chromatin resulting in no loss of H3 or H3K9me3 in ATRX-KO and Control cells between 4 and 8 hpi (*Figure 7D*; *Figure 7—figure supplement 2C*). Taking these results together, we concluded that ATRX plays a role in a process that protects against chromatin destabilization during transcription.

## PML promotes maintenance of viral chromatin during infection

Because PML has recently been implicated in influencing ATRX/DAXX activity (*Delbarre et al., 2017*), we next investigated if PML affects viral chromatin maintenance. We infected HFFs stably expressing a short hairpin against PML (shPML) (*Wagenknecht et al., 2015*) with HSV 7134, and we observed that at 4 hpi, PML-depleted cells (*Figure 8A*) showed no significant differences from shControl cells for H3 but a slight decrease in H3K9me3 at the *ICP27* and *ICP8* viral gene promoters (*Figure 8B–D*). However, by 8 hpi, we observed a significant reduction of chromatin in shPML cells that was similar to what was observed in ATRX-KO fibroblasts (*Figure 8B–D*). Though PML may play a role in promoting early H3K9me3 density at viral gene promoters, these results indicated that PML, like ATRX, is not uniquely required for the formation of viral heterochromatin. However, ATRX and PML are both required for the maintenance of viral heterochromatin stability.

## Discussion

The results presented here argue that viral DNAs are concurrently detected by multiple DNA sensing and restriction pathways and that the initial deposition of chromatin on the HSV genome and the maintenance of viral chromatin are mediated by separate mechanisms. ATRX and IFI16 are both recruited to incoming viral DNA, but they do so independently and then restrict viral infection through independent pathways. Although ATRX has been reported to be important for H3.3 occupation on both cellular and viral genomes, direct examination of the role ATRX plays in de novo heterochromatin formation in vivo has yet to be described. Despite possessing H3.3-specific chaperone activity when in complex with DAXX, here we report that ATRX is dispensable for the initial deposition of histones and repressive histone modifications on naked viral DNA. But ATRX is required to maintain the stability of viral heterochromatin in the face of chromatin stress. This observation reveals new insights into the specificity of histone chaperone activity and may prove to be important in understanding the complex roles ATRX plays in regulating gene expression, heterochromatin maintenance and telomere homeostasis on host cell chromatin.

### Multiple restriction pathways detect incoming viral DNA

Our results revealed that incoming viral DNA is detected and epigenetically silenced by multiple host-cell restriction pathways. We detected IFI16, PML, and ATRX colocalizing with viral DNA by 15 mpi, the earliest time point at which we were able to detect viral DNA in the nuclei of HFFs. This observation argued that these host restriction factors detect HSV almost immediately upon nuclear entry. IFI16 has been observed to form transient foci at the nuclear periphery during early HSV infection (*Diner et al., 2016*; *Everett, 2016*), but there was no definite way to connect the IFI16 foci to incoming viral DNA without labeling of the viral DNA as presented here. IFI16 colocalization with viral DNA was temporally limited, suggesting an initial detection of viral DNA followed by a displacement by additional factors or modifications, perhaps by nucleosomes which are known to impede

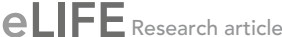

**Figure 8.** PML promotes maintenance of viral chromatin. (A) Immunoblot for PML in shControl and shPML cells. shPML and shControl cells were infected with HSV 7134 at an MOI of 3. Samples were fixed and harvested at 4 and 8 hpi. ChIP-qPCR for (B) H3 and (C) H3K9me3 at the *ICP27* viral gene promoter. ChIP-qPCR for (D) H3 and (E) H3K9me3 at the *ICP8* viral gene promoter. *Figure 8B–E* are reported as the percent of input immunoprecipitated by each antibody and are the average of 3 independent experiments ± standard error of the mean; two-tailed T test, p < 0.05 (*).
DOI: https://doi.org/10.7554/eLife.40228.022

IFI16 oligomerization (*Stratmann et al., 2015*). In contrast, we observed ATRX stably colocalizing with viral genomes until PML-NBs were disbursed by ICP0 around two hpi. We previously showed that depletion of IFI16 does not affect recruitment of PML to ICP4, a marker for viral DNA (*Orzalli et al., 2013*). Furthermore, during the preparation of this manuscript, it was reported that IFI16 does not influence PML recruitment to labeled HSV DNA (*Alandijany et al., 2018*). The latter report supports our own observations that ATRX and IFI16 are independently recruited to incoming viral DNA. Providing additional support for this argument, we showed the restrictive effects of ATRX and IFI16 on ICP0-null HSV replication are additive. These results echo our recent findings regarding the additive restrictive effects of DAXX and IFI16 (*Merkl et al., 2018*). Together, these results argue

that IFI16 and PML-NB components represent distinct nuclear DNA sensing pathways that act in parallel to detect and restrict invading viral DNA almost immediately upon nuclear entry.

## Formation of viral chromatin does not require the ATRX/DAXX complex

Our results revealed that ATRX is not required for H3/H3.3 deposition on viral DNA for the first 4 hpi; however, we cannot rule out the possibility that other histone chaperones or assembly complexes may compensate for the loss of ATRX/DAXX histone chaperone activity, as HIRA does for depletion of CAF-1 (52). In fact, CAF-1 has been reported to compensate for loss of Daxx in murine cells (*Drané et al., 2010*), and NAP1 can assemble H3.3-containing nucleosomes in vitro (*Lewis et al., 2010*). However, it is important to note that ATRX and DAXX have roles in promoting heterochromatin that do not require histone chaperone activity (*Sadic et al., 2015*). Because ATRX can also act as a H3K9me3 reader (*Eustermann et al., 2011*; *Iwase et al., 2011*), restrictive effects of ATRX and DAXX on HSV infection may not require the H3.3 chaperone activity of the ATRX/DAXX complex during lytic infection. Individual and combinatorial depletion of other histone loaders and nucleosome formation complexes will be required to identify the cellular factors needed for the initial loading of heterochromatin on the HSV genome.

We also cannot rule out the possibility that the observed effects are due to an indirect effect as a consequence of depleting ATRX. Depleting cells of a histone chaperone will likely result in changes to cellular chromatin and altered gene expression. However, changes in viral chromatin density in ATRX-KO cells are not observable until after 4 hr post infection, so we do not believe the observed to be effects to be the consequence of enhanced global histone mobility. Additionally, we observed the association of ATRX with viral gene promoters by 4 hr post infection. This finding is consistent a direct role for ATRX in viral chromatin regulation. Furthermore, as both siRNA and CRISPR depletions of ATRX have very similar effects on viral yield and viral gene expression, we believe we can rule out off-target effects.

Our results suggest that DAXX may have ATRX-independent functions during viral infection. Here we report that siRNA depletion of DAXX in ATRX-KO cells significantly decreased the expression of *gB* transcripts during productive ICP0-null HSV infection. Similar dual anti-viral and pro-viral effects have recently been reported for PML (*Newhart et al., 2013*; *Xu et al., 2016*), and the H3.3-specific chaperone, HIRA, has been shown to both enhance and restrict viral infection depending on context (*Placek et al., 2009*; *Rai et al., 2017*). It is possible that DAXX is restrictive while in complex with ATRX but at a later stage of infection, after dispersion of PML-NBs and degradation of ATRX, DAXX may promote HSV gene expression or replication. Our results contribute to a growing body of literature that is revealing ATRX-independent functions of DAXX that may have important implications for viral restriction, cancer biology, and epigenetics (*He et al., 2015*; *Hoelper et al., 2017*).

## ATRX mediates HSV heterochromatin stability during chromatin stress

We hypothesize that ATRX restricts viral infection by maintaining viral heterochromatin stability, resulting in DNA that is less accessible to transcription and viral replication factors (*Figure 9*). Though de novo formation of HSV-associated heterochromatin was unaffected by the absence of ATRX (*Figure 9B*), decreased histone retention or histone reloading in the absence of ATRX/DAXX could conceivably explain the observed loss of H3 and H3K9me3 on HSV genomes, elevated levels of viral transcription, and the enhanced accumulation of nascent viral DNA that we observed by 8 hpi in ATRX-KO cells. Decreased histone retention may lead to increased histone dynamics, reduced nucleosome density, and transiently expose transcription factor binding sites, resulting in elevated transcription (*Huang and Zhu, 2014*) and increased accessibility for viral replication factors. Similarly, if nucleosomes are lost from viral DNA during transcription or DNA synthesis, new nucleosomes will need to be rapidly formed and modified with repressive histone tail modifications to maintain epigenetic silencing. These ideas are consistent with our findings that ATRX promotes the maintenance and continued accumulation of viral heterochromatin, both histones and histone modifications, during times associated with viral transcription and DNA synthesis. When viral DNA synthesis was blocked with PAA, HSV DNA in Control cells continued to accumulate H3 and H3K9me3 from 4 to 8 hpi. In contrast, HSV DNA-associated H3 and H3K9me3 did not accumulate significantly in ATRX-KO cells from 4-8hpi, suggesting that in the absence of ATRX, histone exchange rates had reached an

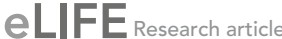

**Figure 9.** Model – ATRX promotes heterochromatin maintenance during challenges to chromatin stability. (**A**) Incoming HSV genomes are rapidly sensed by IFI16, PML, and ATRX upon nuclear entry. (**B**) The de novo formation of heterochromatin on input viral genomes occurs in an ATRX-independent manner, possibly through a HIRA/ASF1A mediated pathway in conjunction with histone methyltransferases (HMTs). However, our results indicate that ATRX is required to maintain viral heterochromatin stability during destabilizing events such as transcription or replication. (**C**) We hypothesize that ATRX maintains viral heterochromatin integrity resulting in reduced chromatin dynamics, stabilized heterochromatin, and reduced access for transcription factors, viral replication factors, and polymerases.

DOI: https://doi.org/10.7554/eLife.40228.023

equilibrium by 4 hpi. While there may be more than one way to explain these observations, we think the model that best fits these observations is one in which ATRX promotes heterochromatin accumulation by mediating modified histone retention.

ATRX is known to be recruited to heterochromatin through the interaction of its ADD domain with the H3K9me3 histone tail modification (*Iwase et al., 2011*), and is known to be required for

both H3.3 and H3K9me3 maintenance on regions of cellular DNA (*Lewis et al., 2010*; *Sadic et al., 2015*; *He et al., 2015*). It can also be recruited by direct interaction with heterochromatin protein 1 (HP1) (*Eustermann et al., 2011*; *Bérubé et al., 2000*), a chromatin remodeling factor which has been reported to localize to HSV gene promoters during infection (*Ferenczy and DeLuca, 2009*). ATRX may act to recruit DAXX to viral DNA through its interaction with H3K9me3 or HP1 and promote rapid replacement of histones that dissociate from viral DNA (*Figure 9C*). These histones could then be rapidly modified with H3K9me3 by histone lysine methyltransferases SUV39H1 or SETDB1 that are known to be recruited by HP1, thus maintaining heterochromatin composition and density on the HSV DNA.

Our model is also likely relevant to the maintenance of DNA virus latency. The EBV protein BNRF1 has been shown to bind to DAXX and separate it from the ATRX/DAXX complex (*Tsai et al., 2011*). That study also showed that depletion of DAXX or ATRX led to EBV reactivation in lymphoblastoid cells. Through its interaction with DAXX, BNRF1 was shown to promote increased mobilization and turnover of H3.3 during FRAP experiments (*Tsai et al., 2014*). The observed increase in H3.3 turnover has been proposed to facilitate expression of viral latent genes and the establishment of latent EBV infection. ATRX and DAXX have also been detected localizing to viral genomes in neurons of mice latently infected with HSV-1 (63). These reports are consistent with our model that ATRX/DAXX promotes heterochromatin integrity to enhance epigenetic silencing of viral genes and proposes that this mechanism may be important for both lytic and latent DNA virus infections. Further studies are needed to define how ATRX reduces the loss of chromatin during times of increased viral transcription and replication.

This model may also be relevant to cellular heterochromatic loci that are maintained by ATRX. Transcription by RNA Pol II evicts histones which are then quickly replaced (*Schwabish and Struhl, 2004*) by the FACT complex, which has been implicated in both the disassembly and reassembly of histone components during transcription (*Belotserkovskaya et al., 2003*; *Mason and Struhl, 2003*; *Xin et al., 2009*). The result is a dynamic equilibrium of nucleosome formation, retention, and eviction. Based on our observations of viral chromatin and viral gene expression in the absence of ATRX, we expect that as the level of transcription increases in the absence of ATRX, the equilibrium between eviction and deposition could shift in favor of eviction and result in an apparent reduction in H3.3 and H3K9me3. Supporting this idea, ATRX has been shown to be required to silence transcription of an eGFP-retrotransposon reporter in response to continuous dox-induced transcription (*Sadic et al., 2015*). In that study, ATRX depletion had minor effects on the initial establishment of H3K9me3 at the reporter element, but it was required for the efficient re-establishment of H3K9me3 after transcriptional challenge was ended. Similarly, ATRX protected against expression of minor satellite sequences in mouse embryonic stem cell derived neurons upon neuronal stimulation (*Noh et al., 2015*).

Like ATRX, PML was not strictly required for the initial deposition of histone H3 or H3K9me3 on viral DNA but required for maintenance of heterochromatin from 4 to 8 hpi. PML was recently reported to influence ATRX/DAXX complex activity and loss of PML may increase H3.3 turnover at PML-associated domains (PADS) (*Delbarre et al., 2017*). Because ATRX is required for heterochromatin maintenance at both pericentric heterochromatin and telomeres (*Lewis et al., 2010*; *Goldberg et al., 2010*; *McDowell et al., 1999*), it is not surprising that pericentric heterochromatin and telomeres have both been shown to exhibit high levels of chromatin stability with very low histone turnover (*Kraushaar et al., 2013*). Consequently, we suggest that a major function of ATRX in heterochromatin maintenance is in stabilizing chromatin density and promoting histone retention. However, increased histone retention is not the only possible mechanism of action. ATRX could also be promoting the re-establishment of heterochromatin in the wake of RNA Pol II mediated transcription, perhaps in a fashion similar to the FACT-complex. Likewise, ATRX could be stabilizing chromatin by inhibiting transcription or replication through a DNA damage pathway. Further mechanistic studies will be required to define the precise mechanism through which ATRX exerts its influence on viral chromatin.

## Epigenetic regulation of viral DNA occurs in two stages

Our finding that ATRX restricts HSV gene expression by promoting the maintenance of viral heterochromatin raises the idea that there are at least two stages of epigenetic regulation involved in restriction of viral gene expression: 1. Initial loading of heterochromatin; 2. Maintenance of

heterochromatin during chromatin stress, such as transcription and replication. Although viral chromatin structure is known to be dynamic (*Gibeault et al., 2016*), before this report, the formation of viral chromatin and the maintenance of viral chromatin were not understood to be separate mechanisms. Importantly, DNA viruses have evolved specific mechanisms for dismantling or disabling the ATRX/DAXX pathway rather than blocking the initial formation of heterochromatin. This argues there may be some role for the early formation of chromatin in the viral life cycle. This is supported by reports that HIRA and ASF1A promote histone occupation on HSV DNA during the first few hours of infection but do not appear to be restrictive during this same period of time (*Oh et al., 2012*; *Placek et al., 2009*). Conversely, the heterochromatin maintenance mechanism mediated by ATRX appears to act as a critical restrictive hurdle to productive infection.

The multiple stages of epigenetic restriction of viral genomes uncovered here involve cellular chromatin factors. Therefore, their roles in epigenetically silencing viral DNA may reflect their functions in cellular chromatin assembly and maintenance. Cellular chromatin structure must be maintained during transcription, DNA synthesis, and mitosis. Thus, it is likely that multiple mechanisms exist to deal with each of these chromatin stressors. The stages of viral epigenetic regulation by the host cell as defined here will serve to identify host epigenetic factors that may function in similar host cell pathways, and this knowledge could provide insights critical to the development of epigenetic therapies for both viral and host diseases.

## Materials and methods

**Key resources table**

| Reagent type (species) or resource | Designation | Source or reference | Identifiers | Additional information |
|---|---|---|---|---|
| Gene (*H. sapiens*) | ATRX | NA | NCBI Gene ID: 546 | |
| Strain, strain background (Herpes simplex virus 1) | KOS | *Smith, 1964* | | |
| Strain, strain background (Herpes simplex virus 1) | HSV-EdC | This Paper | | From HSV KOS |
| Strain, strain background (Herpes simplex virus 1) | HSV 7134 | *Cai and Schaffer, 1992*, PMID: 1313909 | | |
| Strain, strain background (Herpes simplex virus 1) | HSV 7134R | *Cai and Schaffer, 1992*, PMID: 1313910 | | |
| Strain, strain background (Herpes simplex virus 1) | HSV d109 | *Samaniego et al., 1998*; PMID: 9525658 | | |
| Cell line (H. sapiens) | Human Foreskin Fibroblast (HFF) | American Type Culture Collection | SCRC-1041 | |
| Cell line (H. sapiens) | TERT-HF | *Bresnahan and Shenk, 2000*, PMID: 10875924 | | via Robert Kalejta |
| Cell line (H. sapiens) | TERT-HF Control | This paper | | |
| Cell line (H. sapiens) | TERT-HF ATRX-KO | This paper | | |

*Continued on next page*

*Continued*

| Reagent type (species) or resource | Designation | Source or reference | Identifiers | Additional information |
|---|---|---|---|---|
| Cell line (H. sapiens) | 293T | American Type Culture Collection | CRL-3216 | |
| Cell line (H. sapiens) | shControl | *Wagenknecht et al., 2015*, PMID: 26057166 | | |
| Cell line (H. sapiens) | shPML | *Wagenknecht et al., 2015*, PMID: 26057166 | | |
| Cell line (H. sapiens) | U2OS | American Type Culture Collection | HTB-96 | |
| Antibody | Anti-ATRX, rabbit polyclonal | Abcam | ab97508 | 1:500 IF; 1:2000 WB; 6 µg/ChIP rxn |
| Antibody | Anti-IFI16 [IFI-230], mouse monoclonal | Abcam | ab50004 | 1:2500 WB; 1:500 IF |
| Antibody | Anti-PML, rabbit polyclonal | Bethyl | A301-167A | 1:2000 WB |
| Antibody | Anti-DAXX, rabbit polyclonal | Sigma | D7810 | 1:2000 WB |
| Antibody | Anti-ICP4 (58S), mouse monoclonal | *Showalter et al., 1981*, PMID: 6277788 | | 1:2000 WB; 1:500 IF |
| Antibody | Anti-ICP8 (383), rabbit polyclonal | *Knipe et al., 1987*, PMID: 3027360 | | 1:2000 WB; 1:500 IF |
| Antibody | anti-GAPDH [EPR16891], rabbit monoclonal | Abcam | ab181602 | 1:5000 WB |
| Antibody | anti- HSV 1-ICP0, mouse monoclonal | East Coast Bio | H1A207 | 1:5000–10000 IF |
| Antibody | Anti-PML [C7], mouse monoclonal | Abcam | ab96051 | 1:500 IF |
| Antibody | Anti- Histone H3 ChIP Grade, rabbit polyclonal | Abcam | ab1791 | 2.5 µg/ ChIP rxn |
| Antibody | Anti-Histone H3.3, rabbit polyclonal | Millipore | 09–838 | 2.5 µg/ ChIP rxn |
| Antibody | Anti-H3K9me3 ChIP Grade, rabbit polyclonal | Abcam | ab8898 | 2.5 µg/ ChIP rxn |
| Antibody | Anti-Histone H3K27me3, rabbit polyclonal | Active Motif | 39156 | 2.5 µg/ ChIP rxn |
| Antibody | Normal Rabbit IgG, rabbit polyclonal | Millipore | NG1893918 | 2.5 µg/ ChIP rxn (6 µg/ATRX ChIP) |
| Recombinant DNA reagent | lentiCRISPR v2 | Addgene | Plasmid #52961 | |
| Sequence-based reagent | ICP8 cDNA FWD | IDT | | 5'-GTCGTTACCGA GGGCTTCAA |

*Continued on next page*

*Continued*

| Reagent type (species) or resource | Designation | Source or reference | Identifiers | Additional information |
|---|---|---|---|---|
| Sequence-based reagent | ICP8 cDNA REV | IDT | | 5'-GTTACCTTGTCCGAGCCTCC |
| Sequence-based reagent | ICP27 cDNA FWD | IDT | | 5'-GCATCCTTCGTGTTTGTCATT |
| Sequence-based reagent | ICP27 cDNA REV | IDT | | 5'-GCATCTTCTCTCCGACCCCG |
| Sequence-based reagent | gB cDNA FWD | IDT | | 5'-TGTGTACATGTCCCCGTTTTACG |
| Sequence-based reagent | gB cDNA REV | IDT | | 5'-GCGTAGAAGCCGTCAACCT |
| Sequence-based reagent | ICP4 cDNA FWD | IDT | | 5'-GCGTCGTCGAGGTCGT |
| Sequence-based reagent | ICP4 cDNA REV | IDT | | 5'-CGCGGAGACGGAGGAG |
| Sequence-based reagent | Q-RT 18 S F | IDT | | 5'-GCCGCTAGAGGTGAAATTCTTG |
| Sequence-based reagent | Q-RT 18 S R | IDT | | 5'-CTTTCGCTCTGGTCCGTCTT |
| Sequence-based reagent | ICP27 Promoter FWD (ChIP) | IDT | | 5'-ACCCAGCCAGCGTATCCACC |
| Sequence-based reagent | ICP27 Promoter REV (ChIP) | IDT | | 5'-ACACCATAAGTACGTGGC |
| Sequence-based reagent | ICP8 Promoter FWD (ChIP) | IDT | | 5'-GAGACCGGGGTTGGGGAATGAATC |
| Sequence-based reagent | ICP8 Promoter REV (ChIP) | IDT | | 5'-CCCCGGGGGTTGTCTGTGAAGG |
| Sequence-based reagent | GAPDH Promoter FWD (ChIP) | IDT | | 5'-CAGGCGCCCAATACGACCAAAATC |
| Sequence-based reagent | GAPDH Promoter REV (ChIP) | IDT | | 5'-TTCGACAGTCAGTCAGCCGCATCTTCTT |
| Sequence-based reagent | sgATRX 1 F | This paper | | 5'-TCTACGCAACCTTGGTCGAA |
| Sequence-based reagent | sgATRX 1 R | This paper | | 5'-TTCGACCAAGGTTGCGTAGA |
| Sequence-based reagent | sgATRX 2 F | This paper | | 5'-CGAAACTAACAGCTGAACCC |
| Sequence-based reagent | sgATRX 2 R | This paper | | 5'-GGGTTCAGCTGTTAGTTTCG |
| Sequence-based reagent | siNT | Dharmacon | ON-TARGETplus Non-targeting pool D-001810–10 | |
| Sequence-based reagent | siATRX | Dharmacon | ON-TARGETplus ATRX pool D-006524–00 | |
| Sequence-based reagent | siIFI16 | Dharmacon | ON-TARGETplus IFI16 pool D-20004–00 | |
| Sequence-based reagent | siDAXX | Dharmacon | ON-TARGETplus DAXX pool D-004420–10 | |
| Peptide, recombinant protein | Streptavidin-488 | Invitrogen | 532354 | 1:1000 |

*Continued on next page*

*Continued*

| Reagent type (species) or resource | Designation | Source or reference | Identifiers | Additional information |
|---|---|---|---|---|
| Peptide, recombinant protein | RNase A | Thermo Science | EN0531 | |
| Commercial assay or kit | DNase Free Kit | Invitrogen | AM1906 | |
| Commercial assay or kit | RNeasy Kit | Qiagen | 74106 | |
| Commercial assay or kit | QIAquick PCR Purification Kit | Qiagen | 28106 | |
| Commercial assay or kit | High Capacity CDNA Kit | Applied BioSys | 4368814 | |
| Software, algorithm | Foci-detection and colocalization software | This paper - Image and Data Analysis Core (IDAC) at Harvard Medical School; doi: 10.5061/dryad.95fs76f | | MATLAB based package; https://datadryad.org/resource/doi:10.5061/dryad.95fs76f |
| Chemical compound, drug | Picolyl-biotin azide | Click Chemistry Tools | product# 1167 | 10 µM final conc. |
| Chemical compound, drug | sodium ascorbate | Sigma | #a4034 | |
| Chemical compound, drug | CuSO$_4$ | Fisher | C489 | |
| Chemical compound, drug | EdC | Sigma | T511307 | 25 mM stock |
| Chemical compound, drug | Prolong gold | Invitrogen | P36934 | |
| Chemical compound, drug | EM Formaldehyde (IF) | Thermo Science | 2908 | 16% methanol free - 2% working conc. |
| Chemical compound, drug | Formaldehyde (ChIP) | Sigma | F8775 | 1% for ChIP crosslinking |
| Chemical compound, drug | Actinomycin-D | Sigma | A1410 | 5 µg/mL final |
| Chemical compound, drug | α-Amanitin | Sigma | A2263 | 2 µg/mL final |
| Chemical compound, drug | Phosphono Acetate | Sigma | 284270 | 400 µg/mL final |
| Chemical compound, drug | Puromycin | Santa Cruz | CAS 53-79-2 | 5 µg/mL final |
| Chemical compound, drug | Roche Complete protease inhibitor | Sigma-Aldrich | 11697498001 | |
| Chemical compound, drug | Fast-Sybr sybr green (qPCR) | Applied Biosystems | 4385612 | |
| Chemical compound, drug | Lipofectamine RNAiMAX transfection reagent | Invitrogen | 56532 | |

## Cell culture, viruses, and infections

HFF, U2OS, and HEK293T cells were obtained from the American Type Culture Collection (Manassas, VA). hTERT immortalized fibroblasts were a kind gift from Robert Kalejta and originally described in *Bresnahan and Shenk, 2000*. HFFs expressing a short hairpin against PML (shPML)

were a kind gift from Thomas Stamminger (*Wagenknecht et al., 2015*). All cells are regularly tested for the presence of mycoplasma contamination. Cells used in this study were mycoplasma free.

All fibroblast cells and HEK293T cells were maintained in Dulbecco's Modified Eagle's medium (DMEM; Corning, Corning NY) supplemented with 10% (v/v) fetal bovine serum (FBS) in humidified 5% $CO_2$ incubators at 37°C. Cells were serially passaged by trypsin-EDTA (0.05%; Corning) treatment and transfer to fresh media. shPML fibroblasts were maintained under 5 µg/ml puromycin.

HSV-1 KOS was used as the wild type virus in this study. HSV-1 KOS 7134 (*Cai and Schaffer, 1992*) was used for experiments requiring an ICP0-null HSV. HSV-1 7134R is an ICP0⁺ virus. HSV-1 $d$109 does not express HSV immediate-early genes (*Samaniego et al., 1998*). Cells were infected with viruses at the indicated MOI in PBS containing 1% (w/v) glucose and 1% (v/v) bovine calf serum (BCS). Infections were carried out at 37°C with constant agitation with 1 hr adsorption time unless otherwise indicated. After 1 hr, the inoculum was removed and replaced with DMEM supplemented with 1% BCS (DMEV) and then incubated at 37°C until times indicated. ICP0-null viral yields were determined by infecting U2OS cells with serial dilutions of viral lysates collected at indicated times. For experiments testing the effects of viral DNA synthesis, PAA (Sigma-Aldrich; St. Louis, MO) was added to the media (400 µg/ml) with 10 mM HEPES at the time of infection and maintained until time of harvest. Actinomycin-D (5 µg/ml; Sigma) and flavopiridol (1 µM; Selleck Chemicals, Houston, TX) were added to cells 4 hpi and 1 hr prior to infection, respectively, and maintained until time of harvest. Cells were pre-treated with α-amanitin (2 µg/ml; Sigma) for 16 hr prior to infection and maintained until time of harvest. To block viral attachment and entry to cells, heparin was added to the viral inoculum (50 µg/ml), and heparin was maintained for the duration of the infection.

## CRISPR/Cas9-mediated knockout of ATRX

DNA oligos coding encoding sgRNAs specific for ATRX were designed using the Optimized CRISPR Design tool (http://crispr.mit.edu/; Zhang Lab, MIT) and cloned into the lentiCRISPR v2 vector (*Busskamp et al., 2014*). HEK293T cells were grown to 70% confluency in 6-well plates and either lentiCRISPR v2 expressing Cas9 or co-expressing an sgRNA targeting ATRX were co-transfected with the psPAX2 and pVSV-G packaging plasmids using Effectene Transfection Reagent per manufacturer's instructions (Qiagen). hTERT immortalized fibroblasts seeded in a 6-well plate were transduced with supernatant (1:1) harvested from HEK293T cells 48 hr post transfection that had been filtered through a 0.45 µm sterile syringe filter (MilliporeSigma; Burlington, MA). Transduced cells were maintained for two days then placed under puromycin selection (5 µg/ml) and expanded for two weeks.

Oligo sequences for anti-ATRX sgRNA expression: sgATRX 1 F TCTACGCAACCTTGGTCGAA sgATRX 1 R TTCGACCAAGGTTGCGTAGA sgATRX 2 F CGAAACTAACAGCTGAACCC sgATRX 2 R GGGTTCAGCTGTTAGTTTCG

## Preparation of EdC-labeled HSV-1 stocks

HFFs were grown to confluency in T-150 flasks, and then left at confluency for 2–3 days. The confluent HFFs were infected with HSV-1 HSV at a MOI of 10 and incubated at 34°C. At 4 hpi, the medium was replaced with fresh medium containing 0.5 µM of the nucleoside analog, EdC. At 34 hpi, the HSV-infected HFFs were washed twice with 5 mL DMEV per wash, and 20 mL of fresh EdC-free media was added to the cells. The infected HFFs were incubated for an additional 2 hr at 34°C. At 36 hpi, cells were harvested by scraping the flasks in 2.5 mL DMEV and 2.5 mL sterile skim milk. The viral lysate was then sonicated using a Misonix S-4000 probe sonicator on setting 30 for 3 cycles: 30 s on, 30 s off. The HSV-EdC lysate was aliquoted and stored at −80°C. Infection of Vero cells with serially diluted virus was used to determine viral titer via plaque assay.

## Protein depletions

To deplete cells of endogenous proteins, siRNAs were transfected into $1 \times 10^5$ cells plated in a 12-well dish using Lipofectamine RNAiMAX Reagent (Invitrogen). At 48 hr post transfection, cells were re-seeded into assay-appropriate dishes and infected 24 hr later.

siRNAs used in this study: siNT: Dharmacon ON-TARGETplus Non-targeting pool D-001810–10
siATRX: Dharmacon ON-TARGETplus ATRX pool D-006524–00
siIFI16: Dharmacon ON-TARGETplus IFI16 pool D-20004–00

siDAXX: Dharmacon ON-TARGETplus DAXX pool D-004420–10

## Antibodies

The following antibodies were used in this study:

IFI16 Abcam ab50004
PML Bethyl A301-167A (immunoblot only)
PML Abcam ab96051 (immunofluorescence only)
DAXX Sigma D7810 (immunoblot only)
GAPDH Abcam ab8245 (immunoblot only)
ICP8 (74)
ICP4 (75)
ICP0 East Coast Bio H1A207
ATRX Abcam 97508
H3 Abcam ab1791 (ChIP)
H3K9me3 Abcam ab8898 (ChIP)
H3K27me3 Active Motif 39156 (ChIP)
H3.3 Millipore 09–838 (ChIP)
Negative control rabbit IgG Millipore NG1893918 (ChIP)

## Immunoblots

Cells were harvested at times indicated in lithium dodecyl sulfate (LDS) sample buffer and incubated at 95°C for 10 min. Protein samples were run on NuPAGE 4–12% bis-Tris gels (Invitrogen). Proteins were transferred from the gel to nitrocellulose or PVDF membranes for imaging via LI-COR Odyssey imager (Lincoln, NE) or film, respectively. Membranes were blocked in either LI-COR blocking solution or 5% powdered milk-PBS containing 0.1% Tween 20 (PBST) for 1 hr. After blocking, membranes were incubated with primary antibody overnight at 4°C. Membranes were washed 3 times with PBST, incubated in secondary antibody for 1 hr at room temperature (IRDye 800CW Goat anti-Mouse IgG and IRDye 800CW Goat anti-Rabbit IgG for LI-COR; $\alpha$-mouse IgG$^+$HRP and $\alpha$-Rabbit IgG$^+$HRP, Cell Signaling 7076 s and 7074 s, respectively for film). Membranes were washed 3 times in PBST. Nitrocellulose membranes were imaged with a LI-COR Odyssey imager. PVDF membranes were incubated with SuperSignal West Pico chemiluminescent substrate (Thermo Scientific) and exposed to film.

## Immunofluorescence and detection of EdC-labeled HSV-1 genomes

Transfected or infected cells were fixed with 2% methanol-free formaldehyde (ThermoFisher), permeabilized with 0.1% Triton-X 100. Permeabilized cells were incubated with primary antibodies for 30 min at 37°C and washed twice with 0.1% PBST for 5 min each, followed by one wash with PBS. Alexa Fluor 594- and 647-conjugated secondary antibodies (Invitrogen; A11032 and a21246, respectively) and DAPI were incubated with cells for 30 min at 37°C. All antibodies were used at 1:500. The coverslips were washed as described above and either mounted on slides with ProLong Gold antifade reagent (Invitrogen) or incubated with click chemistry reagents as follows. After incubation with secondary antibodies, coverslips were incubated with 10 μM biotin-picolyl azide (Click Chemistry Tools, Scottsdale, AZ), 10 mM sodium ascorbate (Sigma-Aldritch A4034), and 2 mM CuSO$_4$ (Fisher) for 2 hr in the dark. Coverslips were washed twice with 0.1% PBST and once with PBS. Coverslips were then incubated with Alexa Fluor 488-conjugated streptavidin (1:1000; Invitrogen S32354) for 30 min in the dark. Coverslips were washed twice in PBS and mounted as described above. Images were acquired using NIS-Elements AR imaging software (Nikon) controlling a Nikon Ti-E inverted microscope system using a Plan Apo 100x/1.45 objective with a Zila sCMOS camera (Andor) and SPECTRA X light engine (Lumencor) or a Nikon Ti-E inverted spinning disk confocal system using laser lines 488, 514, 561 and 640 and a Plan Apo 100x/1.45 objective. Image J (FIJI) was used to minimally adjust contrast and generate 3D projections of exported images.

## Quantitative image analysis

The Image and Data Analysis Core (IDAC) at Harvard Medical School developed a custom MATLAB-based software for nuclear foci and co-localization detection in microscopy images based upon their

previously described nuclear foci detection software (*Cicconet et al., 2017*; *Cicconet, 2017*). In brief, unaltered images were analyzed by the software to detect signal intensity over a set threshold to define foci above background. The software then uses DAPI staining to generate a mask that defines nuclear areas in the image. Only foci above the fluorescence threshold and within the nuclear masks were scored. The software determines colocalization based on the distance between the centers of nuclear foci in a reference channel (vDNA) to foci in other channels of the image. The centers of compared foci must be ≤ 5 pixels (~350 nm at approximately 70 nm per pixel) in distance to be considered colocalized (user definable). An additional MATLAB script uses nuclear masks and foci center x,y coordinate data from each channel to determine distances of nearest neighbor foci in a second channel from foci in the reference channel and generate distance frequency data (*Cicconet, 2018*). The script then generated the same number of random x,y points within the nuclear mask as were detected in the second channel. The distances from the centers of reference channel foci to nearest neighbor random points was then calculated and reported to within a user defined radius (60 pixel radius for data in this paper) of each reference focus. The resulting frequency distributions of reference-to-test channel distances and reference-to-random points were compared by the non-parametric Kolmogorov-Smirnov test. If the test rejects the null hypothesis at 5% significance, the test returns a value of 1, and 0 if the null hypothesis is not rejected. The test also returns an asymptotic p-value and a list of distances in pixel values which can then be plotted and further analyzed using GraphPad Prism (as it was here) or other data analysis packages. Foci detection and colocalization software and nearest neighbor analysis script are available for use: doi:10.5061/dryad.95fs76f.

## RT-qPCR quantification of viral transcripts

Total RNA was isolated using the Qiagen RNeasy kit per manufacturer's instructions. Resulting RNA was quantified, treated with DNAse (DNAfree Kit, Ambion), and reverse transcribed (High Capacity cDNA RT Kit, Applied Biosystems) to produce cDNA. Quantitative PCR was performed using Fast SYBR Green reagents (ThermoFisher) on a StepOnePlus from Applied Biosystems (ThermoFisher). Oligos for qPCR reactions are as follows:

ICP27 (mRNA) for GCATCCTTCGTGTTTGTCATT
ICP27 (mRNA) rev GCATCTTCTCTCCGACCCCG
ICP8 (mRNA) for GTCGTTACCGAGGGCTTCAA
ICP8 (mRNA) rev GTTACCTTGTCCGAGCCTCC gB (mRNA) for TGTGTACATGTCCCCG TTTTACG gB (mRNA) rev GCGTAGAAGCCGTCAACCT
18S (mRNA) for GCCGCTAGAGGTGAAATTCTTG
18S (mRNA) rev CTTTCGCTCTGGTCCGTCTT

## Chromatin immunoprecipitation

ChIP experiments were carried out as previously described in detail (*Lee et al., 2016*) with minor changes. Briefly, infected cell monolayers were fixed in 1% formaldehyde (Sigma-Aldrich) at 37°C for 15 min. Fixation was quenched with glycine at a final concentration of 125 mM for 3 min. Cells were washed 3 times in ice cold PBS supplemented with 1 mM phenylmethanesulfonylfluoride (PMSF), collected in PBS + PMSF, and pelleted at 2000 rpm for 5 min at 4°C. Cell pellets were re-suspended in 1 mL lysis buffer (1% SDS, 10 mM EDTA, 50 mM Tris, pH 8.1), transferred to 15 mL polystyrene tubes and sonicated at 4°C in a Diagenode Biorupter on high setting for 9 cycles of 5 min each (30 s ON, 30 s OFF). An aliquot of recovered chromatin was used for each IP reaction in 1 ml IP dilution buffer (150 mM NaCl, 10 mM $Na_2HPO_4$, 2 mM EDTA, 1.1% Triton, 0.1% SDS). An aliquot of a no antibody IP reaction was retained as 10% input sample. IPs were performed with 2.5 µg (6.0 µg for ATRX ab) of ChIP grade antibody (see Antibodies) per reaction and incubated overnight at 4°C. 20 µl of MagnaChIP protein A magnetic beads (Millipore) were added for 2.5–3 hr at 4°C with rotation, washed 3 times with a cold low-salt buffer (150 mM NaCl, 20 mM Tris·HCl, pH 8.1, 2 mM EDTA, 1% Triton X-100, and 0.1% SDS, 1 mM PMSF) and 3 times with cold LiCl wash buffer (50 mM HEPES pH 7.5, 250 mM lithium chloride, 1 mM EDTA, 1% NP-40, 0.7% sodium deoxycholate, 1 mM PMSF), and washed once with cold Tris-EDTA buffer (10 mM Tris-HCl, pH 8, 1 mM EDTA). DNA-protein complexes were eluted in 100 µl of elution buffer (1% SDS, 0.1 M $NaHCO_3$) at 65°C for 20 min. Protein-DNA crosslinks were reversed by adding NaCl (0.2 M final concentration) and incubating at 95°C for

30 min followed by 1 hr RNase A (Ambion) treatment at 37°C and Proteinase K treatment (Roche) at 45°C for 2 hr. DNA was purified using a QIAquick PCR purification kit (Qiagen) end eluted twice with 50 µl buffer EB for a final volume of 100 µl.

Quantitative-PCR was performed to quantify DNA using Fast SYBR Green reagents (Thermo-Fisher) on a StepOnePlus from Applied Biosystems (ThermoFisher). Oligos for qPCR reactions are as follows:

*ICP27* promoter (genomic)
FWD ACCCAGCCAGCGTATCCACC
  REV ACACCATAAGTACGTGGC

*ICP8* promoter (genomic)
FWD GAGACCGGGGTTGGGGAATGAATC
  REV CCCCGGGGGTTGTCTGTGAAGG

*GAPDH* promoter (genomic)
FWD CAGGCGCCCAATACGACCAAAATC
  REV TTCGACAGTCAGTCAGCCGCATCTTCTT

## Acknowledgments

This work was supported by NIH grant AI106934. We thank the Microscopy Resources on the North Quad (MicRoN) core at Harvard Medical School for consultation and assistance during image acquisition. We thank the Image and Data Analysis Core (IDAC) at Harvard Medical School for consultation and coding of the foci detection and colocalization software and frequency distribution script used in this study. We thank Jeho Shin for technical assistance and Patrick T. Waters for assistance with the manuscript.

## Additional information

### Competing interests

David M Knipe: Reviewing editor, *eLife*. The other authors declare that no competing interests exist.

### Funding

| Funder | Grant reference number | Author |
| --- | --- | --- |
| National Institutes of Health | AI106934 | David M Knipe |

The funders had no role in study design, data collection and interpretation, or the decision to submit the work for publication.

### Author contributions

Joseph M Cabral, Conceptualization, Formal analysis, Validation, Investigation, Visualization, Methodology, Writing—original draft, Project administration, Writing—review and editing; Hyung Suk Oh, Methodology, Significant contributions to the generation of EdC labeled HSV and to the generation of CRISPR-Cas9 knock out cell lines; David M Knipe, Conceptualization, Supervision, Funding acquisition, Methodology, Writing—original draft, Project administration, Writing—review and editing

### Author ORCIDs

Joseph M Cabral http://orcid.org/0000-0003-4733-6612
Hyung Suk Oh http://orcid.org/0000-0002-1739-0389
David M Knipe http://orcid.org/0000-0003-1554-6236

Decision letter and Author response
Decision letter https://doi.org/10.7554/eLife.40228.028
Author response https://doi.org/10.7554/eLife.40228.029

## Additional files

### Supplementary files

• Transparent reporting form
DOI: https://doi.org/10.7554/eLife.40228.024

### Data availability

All data generated or analyzed during this study are included in the manuscript and supporting files. MATLAB based software developed to generate image analysis data is available for use: doi:10.5061/dryad.95fs76f

The following dataset was generated:

| Author(s) | Year | Dataset title | Dataset URL | Database and Identifier |
|---|---|---|---|---|
| Cabral J, Oh H, Knipe D | 2018 | Data from: ATRX promotes maintenance of herpes simplex virus heterochromatin during chromatin stress | https://dx.doi.org/10.5061/dryad.95fs76f | Dryad Digital Repository, 10.5061/dryad.95fs76f |

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
