## [Decision Letter]

Thank you for submitting your article "ATRX promotes maintenance of herpes simplex virus heterochromatin during chromatin stress" for consideration by *eLife*. Your article has been reviewed by three peer reviewers, and the evaluation has been overseen by a Reviewing Editor and Jessica Tyler as the Senior Editor. The following individual involved in review of your submission has agreed to reveal their identity: Luis Schang (Reviewer #3).

The reviewers have discussed the reviews with one another and the Reviewing Editor has drafted this decision to help you prepare a revised submission.

Summary:

The authors are investigating the recruitment and functional roles of host factors that localize with HSV DNA early after infection. Using click chemistry, they are able to defect HSV DNA in the nucleus, and thereby follow the earliest recruitment host factors. The focus of the work is to understand the mechanisms by which host factors recognize, and epigenetically restrict viral DNAs. The authors find that the DNA sensor IFI16, PML, and the histone chaperone ATRX are recruited to viral DNA within 15 minutes post infection. The authors investigated possible sequential recruitment among these three factors, and then focused on ATRX. The HSV genome encodes a protein (ICP0) that disables ATRX, suggesting that ATRX is an antiviral factor. To test this directly, the authors used ATRX knockout cells, and an HSV mutant lacking ICP0. They find that viral transcripts increased in response to ATRX knockout even when viral DNA amplification was blocked. Although ATRX was reported to be a histone chaperone capable of loading histone H3.3 on viral DNA, the authors could not detect early differences in viral chromatin formation. However, several hours later, reduced levels of generic and repressive chromatin were observed in the absence of ATRX. These results pointed to a role of ATRX in maintaining, rather than initiating repressive chromatin. Additional experiments indicate that ATRX is important in stabilization of viral chromatin and heterochromatin in the face of chromatin stress events that occur during viral replication, namely viral DNA replication and RNA transcription.

Essential revisions:

We include here the key points raised by the three reviewers. We judge that their major points can be addressed by some limited experimental work and some rewriting.

*Reviewer #1:*

The experiments are generally well designed and readouts are appropriately quantitative. The overall findings are limited largely to ATRX, while the overall manuscript is expansive and covers many other details.

One experimental design concern relates to the host cell status. In measuring early events through such short timescales, it is surprising that there is no consideration of cell cycle, which might be expected to impact a variety of host-virus interactions. The events being monitored are presumably occurring prior to viral gene expression and manipulation of the cell cycle by HSV. It would seem that host cell cycle status could be important for interpreting the findings. For example, ATRX is known to be cell cycle regulated.

*Reviewer #2:*

This paper presents interesting new data about the epigenetic restriction of herpes viral DNAs in the early times after their nuclear entry. The data show that ATRX is needed to maximally repress viral DNA expression and replication. ATRX KO cells support more robust virus expression and replication. siRNA KD of DAXX did not further relieve repression of ATRX KO cells, suggesting that they work together.

The key new information is that viral DNA forms heterochromatin even without ATRX, at early times; but the heterochromatin is unstable without ATRX at later times. Thus, there are two phases to the story. Inhibiting transcription/replication rescues the chromatin loss (really heterochromatin loss?) – the simplest interpretation is that chromatin loss (really heterochromatin loss?) is mediated by transcription- or replication-induced stress. (Other interpretations are possible).

The basic idea of two phases seems clear and important. I think we do not quite know what is going on when the histone marks are lost vs. maintained, and this is not explained adequately.

Specific comments:

Often in the paper a particular factor is described as "required for the maintenance of viral chromatin". I think this is unclear: one reading of this is that chromatin per se is not maintained – that nucleosomes are removed. Maybe what is meant is that histone marks are lost – and that the factor is "required for the maintenance of viral heterochromatin", or maybe for the "state of the chromatin", or maybe for "histone isoforms and histone marks". If what is meant is histone loss per se, this needs to be clarified. At very late times (8 h post infection) is the idea that there are no histones on the viral DNA? Are viral DNAs associating with other proteins that will coat the DNA in the virion?

The experiments for this, in any case, are ChIP readouts of H3, H3.3, H3K9me3, and H3K27me3. So could not these data address this issue of order of events? Are we seeing loss of all H3, or H3.3, or just loss of marked H3? Can we extract information from the kinetics (Figure 5)? Maybe one cannot say too much. There is some discussion of the ideas of histone loss and replacement and remarking during transcription and replication (subsection “ATRX mediates HSV heterochromatin stability during chromatin stress”) but it is not clear what the authors really think is going on.

Comment on factors on DNA:

The data for the colocalization by IF of viral DNA with ATRX and IFI16, are fine, though the proximity needed for scoring here is probably not necessarily intimate. (The fact that they both become associated with similar kinetics probably does not warrant the claim "that ATRX localizes to viral DNA through PML-NBs and its interaction with DAXX". Not safe; not needed.) I do find the IF assay rather weak.

Would ChIP-seq work to study the association? Probably not, if there are no specific binding sites. But this should be discussed, if not tried.

*Reviewer #3:*

The manuscript presents mostly through analyses of the roles of the proteins tested, using state of the art and more established approaches. It helps clarifying our current understanding of the silencing of the infecting genomes and explores mechanisms of action. The experiments are appropriate to test the proposed models, and the results are mostly properly presented. The interpretation of some of the results does not appear to be supported by appropriate statistical analyses, and the presentation of some of the results could be improved. The general description of the results through the paper does not come across as in a dynamic context for the viral chromatin (dynamics of cellular chromatin are, in contrast, explicitly discussed). The main points are discussed below.

The authors state, in the second paragraph of the subsection “HSV genomes colocalize with host restriction factors IFI16, ATRX, and PML upon entry into the nucleus”, that IFI16 and ATRX are among the first cellular proteins to respond to nuclear entry of viral DNA. This conclusion is not really supported. Only one other protein is really evaluated, PML, and it localizes with the viral genomes with the same kinetics as ATRX. In the same page, the authors state that the expression of ICP0 was inversely correlated to the presence of ATRX foci, which is correct but only in the most general terms. There is little change in ATRX foci between 60 and 90 minutes when the cells expressing ICP0 increase from less than 20 to around 60%, and there is a decrease from almost 100% to around 40% of cells showing ATRX foci between 90 and 120 minutes, when the percentage of cells expressing ICP0 does not change much. The text should be edited to better discuss these results.

The comparison in Figure 4B-C between the effects of knocking down DAXX or IFI16 in combination with ATRX knockout is not well supported. Different MOIs were used in each analysis (3 for DAAX in combination with ATRX and 0.1 for the combination of IFI16 and ATRX). These experiments should be repeated at the same MOI.

The final model presented in Figure 9, and the text in general, suggests that the authors consider the viral chromatin to be dynamic. However, the word dynamic is used only three times, to discuss either chromatin in general or the cellular chromatin in particular. If the authors indeed think that the viral chromatin is dynamic (as it appears to be), then they may want to clarify it through the text. For example, the last paragraph of the subsection “ATRX mediates HSV heterochromatin stability during chromatin stress”, mentions a function in stabilizing chromatin density and promoting histone retention, which appears to indicate decreasing viral chromatin dynamics. The authors also state, "before this report epigenetic silencing of incoming viral genomes was thought to be largely the result of the initial formation of viral heterochromatin." However, several published papers refer to the dynamics of the viral chromatin as important in regulating HSV-1 gene expression.

Knocking out, or even knocking down, histone chaperones would be expected to also affect cellular chromatin, which would be predicted to affect the dynamics of the histones (and all other chaperones) and may very well affect cellular gene expression. The paper would benefit from some discussion of this unavoidable and likely effect of the use of knockouts or knockdowns.

---

## [Author Response]

Reviewer #1:The experiments are generally well designed and readouts are appropriately quantitative. The overall findings are limited largely to ATRX, while the overall manuscript is expansive and covers many other details.One experimental design concern relates to the host cell status. In measuring early events through such short timescales, it is surprising that there is no consideration of cell cycle, which might be expected to impact a variety of host-virus interactions. The events being monitored are presumably occurring prior to viral gene expression and manipulation of the cell cycle by HSV. It would seem that host cell cycle status could be important for interpreting the findings. For example, ATRX is known to be cell cycle regulated.

ATRX shows changes during mitosis only, and mitosis is the only stage of the cell cycle where HSV shows any reductions in infection (Drayman et al., 2017). In fact, HSV also blocks the cell cycle in G1 or G2, so there are not many infected cells in M that would be scoring in our analysis and therefore not likely to be affecting our results. We feel that discussing this would likely add too much complexity to the manuscript for a likely small effect.

Reviewer #2:This paper presents interesting new data about the epigenetic restriction of herpes viral DNAs in the early times after their nuclear entry. The data show that ATRX is needed to maximally repress viral DNA expression and replication. ATRX KO cells support more robust virus expression and replication. siRNA KD of DAXX did not further relieve repression of ATRX KO cells, suggesting that they work together.The key new information is that viral DNA forms heterochromatin even without ATRX, at early times; but the heterochromatin is unstable without ATRX at later times. Thus, there are two phases to the story. Inhibiting transcription/replication rescues the chromatin loss (really heterochromatin loss?) – the simplest interpretation is that chromatin loss (really heterochromatin loss?) is mediated by transcription- or replication-induced stress. (Other interpretations are possible).The basic idea of two phases seems clear and important. I think we do not quite know what is going on when the histone marks are lost vs. maintained, and this is not explained adequately.As described below, we have clarified the wording on the chromatin.Specific comments:Often in the paper a particular factor is described as "required for the maintenance of viral chromatin". I think this is unclear: one reading of this is that chromatin per se is not maintained – that nucleosomes are removed. Maybe what is meant is that histone marks are lost – and that the factor is "required for the maintenance of viral heterochromatin", or maybe for the "state of the chromatin", or maybe for "histone isoforms and histone marks". If what is meant is histone loss per se, this needs to be clarified. At very late times (8 h post infection) is the idea that there are no histones on the viral DNA? Are viral DNAs associating with other proteins that will coat the DNA in the virion?

We agree that we were imprecise in our wording in certain places, and we have revised the text to clearly state that ATRX plays a role in maintaining heterochromatin in the form of both nucleosomes and heterochromatin marks, but a greater removal of H3K9me3 modification.

The experiments for this, in any case, are ChIP readouts of H3, H3.3, H3K9me3, and H3K27me3. So could not these data address this issue of order of events? Are we seeing loss of all H3, or H3.3, or just loss of marked H3? Can we extract information from the kinetics (Figure 5)? Maybe one cannot say too much. There is some discussion of the ideas of histone loss and replacement and remarking during transcription and replication (subsection “ATRX mediates HSV heterochromatin stability during chromatin stress”) but it is not clear what the authors really think is going on.

We have not been able to distinguish the kinetics of the loss of nucleosomes from heterochromatin marks. We have clarified the language of our model as described above.

Comment on factors on DNA:The data for the colocalization by IF of viral DNA with ATRX and IFI16, are fine, though the proximity needed for scoring here is probably not necessarily intimate. (The fact that they both become associated with similar kinetics probably does not warrant the claim "that ATRX localizes to viral DNA through PML-NBs and its interaction with DAXX". Not safe; not needed.) I do find the IF assay rather weak.Would ChIP-seq work to study the association? Probably not, if there are no specific binding sites. But this should be discussed, if not tried.To address the concern regarding interaction with the viral DNA, we have added experiments (Figure 3A) in which we use an ATRX antibody to perform a ChIP assay that detected ATRX at viral gene promoters by 4hours post infection in HFFs.Reviewer #3:The manuscript presents mostly through analyses of the roles of the proteins tested, using state of the art and more established approaches. It helps clarifying our current understanding of the silencing of the infecting genomes and explores mechanisms of action. The experiments are appropriate to test the proposed models, and the results are mostly properly presented. The interpretation of some of the results does not appear to be supported by appropriate statistical analyses, and the presentation of some of the results could be improved. The general description of the results through the paper does not come across as in a dynamic context for the viral chromatin (dynamics of cellular chromatin are, in contrast, explicitly discussed). The main points are discussed below.The authors state, in the second paragraph of the subsection “HSV genomes colocalize with host restriction factors IFI16, ATRX, and PML upon entry into the nucleus”, that IFI16 and ATRX are among the first cellular proteins to respond to nuclear entry of viral DNA. This conclusion is not really supported. Only one other protein is really evaluated, PML, and it localizes with the viral genomes with the same kinetics as ATRX.

We agree and have modified the text to merely state that these proteins co-localize with viral DNA soon after nuclear entry of the DNA.

In the same page, the authors state that the expression of ICP0 was inversely correlated to the presence of ATRX foci, which is correct but only in the most general terms. There is little change in ATRX foci between 60 and 90 minutes when the cells expressing ICP0 increase from less than 20 to around 60%, and there is a decrease from almost 100% to around 40% of cells showing ATRX foci between 90 and 120 minutes, when the percentage of cells expressing ICP0 does not change much. The text should be edited to better discuss these results.

We have edited the text to further discuss these observations in more detail. While the high background from ICP0 antibody staining made it technically difficult to perform precise colocalization studies, in general, the period between 60-90 minutes post infection saw a dramatic rise in ICP0 and ATRX colocalization. The period of time between 90-120 minutes may reflect an increase in ICP0 accumulation at PML-NBs followed by the disruption of these bodies. Between 120-150 minutes post infection, ICP0 and ATRX do appear more closely inversely correlated.

The comparison in Figure 4B-C between the effects of knocking down DAXX or IFI16 in combination with ATRX knockout is not well supported. Different MOIs were used in each analysis (3 for DAAX in combination with ATRX and 0.1 for the combination of IFI16 and ATRX). These experiments should be repeated at the same MOI.

We have repeated the experiment in Figure 4B at MOI 0.1 to match the experimental conditions in Figure 4C. The previous Figure 4B panel has been moved to the supplement. The observations remained the same, and the text was modified to reflect the additional conditions.

The final model presented in Figure 9, and the text in general, suggests that the authors consider the viral chromatin to be dynamic. However, the word dynamic is used only three times, to discuss either chromatin in general or the cellular chromatin in particular. If the authors indeed think that the viral chromatin is dynamic (as it appears to be), then they may want to clarify it through the text. For example, the last paragraph of the subsection “ATRX mediates HSV heterochromatin stability during chromatin stress”, mentions a function in stabilizing chromatin density and promoting histone retention, which appears to indicate decreasing viral chromatin dynamics. The authors also state, "before this report epigenetic silencing of incoming viral genomes was thought to be largely the result of the initial formation of viral heterochromatin." However, several published papers refer to the dynamics of the viral chromatin as important in regulating HSV-1 gene expression.

We do believe that HSV chromatin, like cellular chromatin, is dynamic. While we did not explicitly use the word dynamic to describe the state of HSV chromatin, we do discuss the ramifications of decreased chromatin stability and the likely increase in histone turnover. The Discussion has been modified to cite a paper by Schang et al. on chromatin dynamics during infection and to reflect this position.

Knocking out, or even knocking down, histone chaperones would be expected to also affect cellular chromatin, which would be predicted to affect the dynamics of the histones (and all other chaperones) and may very well affect cellular gene expression. The paper would benefit from some discussion of this unavoidable and likely effect of the use of knockouts or knockdowns.

We agree with this assessment and have added a brief discussion to the text to address these unavoidable concerns. We have also added Figure 3A showing association of ATRX with viral DNA, adding evidence for a direct effect.